# Transcriptomics of *Besnoitia besnoiti*-Infected Fibroblasts Reveals Hallmarks of Early Fibrosis and Cancer Progression

**DOI:** 10.3390/microorganisms12030586

**Published:** 2024-03-15

**Authors:** María Fernández-Álvarez, Pilar Horcajo, Alejandro Jiménez-Meléndez, Pablo Angulo Lara, Ana Huertas-López, Francisco Huertas-López, Ignacio Ferre, Luis Miguel Ortega-Mora, Gema Álvarez-García

**Affiliations:** 1SALUVET, Animal Health Department, Faculty of Veterinary Sciences, Complutense University of Madrid, 28040 Madrid, Spain; marfer23@ucm.es (M.F.-Á.); phorcajo@ucm.es (P.H.); ajmelendez@ucm.es (A.J.-M.); pabang01@ucm.es (P.A.L.); ana.huertas@um.es (A.H.-L.); iferrepe@ucm.es (I.F.); luis.ortega@ucm.es (L.M.O.-M.); 2Animal Health Department, Faculty of Veterinary Sciences, University of Murcia—Regional Campus of International Excellence “Campus Mare Nostrum”, 30100 Murcia, Spain; 3Marbyt—Smart Solutions for Biotechnology, 30100 Murcia, Spain; francisco.huertas@marbyt.com

**Keywords:** *Besnoitia besnoiti*, bovine aorta fibroblasts, RNA-Seq, early fibrosis, MAPK signaling, TGF*β*, cancer progression

## Abstract

Endothelial injury, inflammatory infiltrate and fibrosis are the predominant lesions in the testis of bulls with besnoitiosis that may result in sterility. Moreover, fibroblasts, which are key players in fibrosis, are parasite target cells in a *Besnoitia besnoiti* chronic infection. This study aimed to decipher the molecular basis that underlies a drift toward fibrosis during the disease progression. Transcriptomic analysis was developed at two times post-infection (p.i.), representative of invasion (12 h p.i.) and intracellular proliferation (32 h p.i.), in primary bovine aorta fibroblasts infected with *B. besnoiti* tachyzoites. Once the enriched host pathways were identified, we studied the expression of selected differentially expressed genes (DEGs) in the scrotal skin of sterile infected bulls. Functional enrichment analyses of DEGs revealed shared hallmarks of cancer and early fibrosis. Biomarkers of inflammation, angiogenesis, cancer, and MAPK signaling stood out at 12 h p.i. At 32 h p.i., again MAPK and cancer pathways were enriched together with the PI3K–AKT pathway related to cell proliferation. Some DEGs were also regulated in the skin samples of naturally infected bulls (*PLAUR*, *TGFβ1*, *FOSB*). We have identified potential biomarkers and host pathways regulated during fibrosis that may hold prognostic significance and could emerge as potential therapeutic targets.

## 1. Introduction

*Besnoitia besnoiti* belongs to the Sarcocystidae family, together with the closely related apicomplexan parasites *Toxoplasma gondii* and *Neospora caninum*. It is the etiologic agent of bovine besnoitiosis, a relevant disease in cattle in Africa, Europe, the Middle East and Asia [1]. The clinical progression of bovine besnoitiosis occurs in a two-stage process. In the initial stage, there is a rapid replication of tachyzoites, which invade the endothelial cells and macrophages. First, infected cattle develop fever, depression and anorexia. Next, this febrile phase is followed by the anasarca stage, characterized by subcutaneous edema, nasal and ocular discharge, respiratory disorders and orchitis. In the scleroderma stage, the tachyzoites transform into slowly dividing bradyzoites that form tissue cysts within fibroblasts and myofibroblasts, primarily in subcutaneous tissues, gaining access to a protective niche capable of avoiding immune clearance. These cysts are commonly observed in chronically infected animals, causing skin lesions such as hyperkeratosis, folding, hair loss, nodules and scars. Additionally, visible cysts can be found in the conjunctiva and *vestibulum vaginae* [2,3]. This parasitic disease impairs reproductive parameters since orchitis may result in sterility [4,5]. The severity of clinical signs and lesions can vary among affected animals. Most animals remain subclinically infected, while a small proportion develop pronounced clinical manifestations and lesions, which can ultimately lead to irreversible lesions and even mortality [3,6]. The unknown *B. besnoiti* complete life cycle, together with the complex pathogenesis of besnoitiosis, is a challenge for the development of effective treatment and control strategies. Therefore, nowadays, control programs are limited to management practices coupled with diagnostic tools to prevent new cases and disease dissemination [7].

To decipher the intricate interplay between the parasite and the host, it is essential to focus on the target cells involved in the infection process. Among other target cells, fibroblasts garner significant attention during *B. besnoiti* infection due to their diverse functions in tissue repair, remodeling and immune responses [8,9]. Moreover, fibroblasts are widely acknowledged as the primary cell type in connective tissue, known for their remarkable versatility in producing a diverse array of vital substances such as collagen proteoglycans, fibronectin, laminins, glycosaminoglycans, metalloproteinases and prostaglandins. Therefore, fibroblasts are responsible for synthesizing and reorganizing the extracellular matrix (ECM) [10]. Additionally, in the context of fibrosis, activated fibroblasts, also known as myofibroblasts, are the key players. These myofibroblasts exhibit enhanced contractile properties and increase the production of ECM components, particularly collagen. They also release profibrotic factors that stimulate further fibroblast activation and collagen deposition, perpetuating the fibrotic process. This leads to functional impairment and can ultimately result in organ failure in severe cases [11,12]. In bovine besnoitiosis, fibrosis is a prominent lesion observed in the scrotal skin of acutely and chronically infected bulls [4,5], and two fibrosis biomarkers (*ICAM-1* and *PLAT*) are upregulated in the testicular parenchyma, pampiniform plexus and scrotal skin of naturally chronically infected bulls [5]. Additionally, evidence of fibrosis occurrence in primary bovine aorta endothelial cells (BAEC) during *B. besnoiti* infection was demonstrated in vitro through RNA-Seq analysis [13].

Nevertheless, there is no available data regarding the molecular mechanisms underlying fibroblast infection and the progression of tissue fibrosis during bovine besnoitiosis. Herein, we analyzed the molecular mechanisms that govern *B. besnoiti* infection in bovine fibroblasts following a transcriptomic approach. RNA-Seq analysis of *B. besnoiti*-infected bovine fibroblasts was carried out at two different time points representing early invasion (12 h p.i.) and intracellular proliferation (32 h p.i.). Next, we analyzed the expression of some selected DEGs in the scrotal skin of naturally infected bulls.

## 2. Materials and Methods

### 2.1. Cell Line and Parasite Culture

Primary bovine aorta fibroblasts [14] were cultured in T25 culture flasks with Dulbecco’s Modified Eagle Medium containing 15% fetal bovine serum and 100 IU/mL of penicillin, 100 mg/mL of streptomycin and 0.25 μg/mL of amphotericin B. Low passage fibroblasts (passage 10) were passaged once a week using pre-mix trypsin EDTA (TrypLE Gibco, Thermo Fisher Scientific, Waltham, MA, USA).

Tachyzoites from the *B. besnoiti* Spain1 isolate (BbSp1) were maintained in African green monkey kidney epithelial cell line MARC-145 cells, following a previously described procedure [15]. Only low passage (10 to 21) tachyzoites were used to avoid changes associated with adaptation to long-term cell culture maintenance [16]. After three days of infection, when most parasites were still intracellular, the tachyzoites were scraped and purified using PD-10 columns. The number of viable tachyzoites was estimated using a trypan blue dye exclusion assay, and the viable tachyzoites were counted on a Neubauer chamber. Purified and viable tachyzoites were used to infect T25 flasks seeded with confluent fibroblasts monolayers (2 × 10^6^ cells) with a multiplicity of infection (MOI) of 10:1. The cell cultures were incubated at 37 °C with 5% CO_2_ in a humidified incubator.

### 2.2. Transcriptome Analysis

#### 2.2.1. Experimental Design and RNA Extraction

Samples were collected at 12 h p.i., when most of the parasites had already invaded the host cell and had not replicated yet (FI-Bb 12 h p.i.), and at 32 h p.i., when the parasites had replicated twice (FI-Bb 32 h p.i.) [14]. Extracellular parasites were eliminated by washing infected flasks with phosphate-buffered saline 1x before the cells were recovered at both time points. Non-infected fibroblasts (FI) were used as a control. The cells were recovered by gentle scraping, centrifuged at 1350× *g* for 10 min at 4 °C, and, once the supernatant was discharged, the pelleted cells were stored at −80 °C until RNA extraction. All the analyses were performed with three biological replicates.

Total RNA from the three independent biological replicates was purified by using a QIAGEN RNeasy Mini Kit (Qiagen, Hilden, Germany) following QIAshredder (Qiagen, Hilden, Germany) homogenization according to the manufacturer’s instructions. RNA integrity was evaluated by 1% agarose gel electrophoresis with GelRed staining (Biotium, Fremont, CA, USA).

#### 2.2.2. Quality Control of Total RNA, Library Preparation and Sequencing

The total RNA’s quality and quantity were assessed using Bioanalyzer 2100 (Agilent Biotek, Santa Clara, CA, USA) and a Qubit 2.0. B (Invitrogen, Carlsbad, CA, USA). Subsequently, the poly(A)+ mRNA fraction was extracted from the total RNA, and cDNA libraries were prepared following Illumina’s guidelines [17]. The quantification of the libraries was performed by qPCR using a LightCycler 480 (Roche, Basel, Switzerland), and their quality was assessed using a Bioanalyzer 2100 (Agilent Biotek, Santa Clara, CA, USA). The sequencing of the equimolarly pooled cDNA libraries was performed by paired-end sequencing (100 bp × 2) using an Illumina HiSeq 2000 sequencer (Illumina, San Diego, CA, USA).

#### 2.2.3. Computational Analysis of RNA-Seq Data

The FastQC tool was used to assess the raw data quality, and any reads of low quality were removed through the utilization of Picard Tools software, version 1.129 (http://picard.sourceforge.net, accessed on 10 February 2023).

The raw paired-end reads were mapped against the *Bos taurus* genome, version UDM3.1 (NCBI:GCA_000003055.3), provided by the ENSEMBL/NCBI database (http://www.ensembl.org/, accessed on 10 February 2023) using the TopHat2 v2.1.1 algorithm [18]. Gene quantification was carried out using htseq_count 0.10 [19]. Differential expression between fibroblasts infected with tachyzoites of *B. besnoiti* (FI-Bb) at 12 h vs. uninfected fibroblasts (FI) and FI-Bb at 32 h vs. FI was studied using the algorithm proposed by DESeq2 [20], with a binomial negative distribution for determination of the statistical significance [21]. Genes were considered differentially expressed when they presented a fold change (FC) ≥ 1.5 and a false discovery rate (FDR)-adjusted [22] *p*-value (p adj) less than 0.05, following previous studies [13,23].

Furthermore, to explore the correlation among all replicates, a principal component analysis (PCA) was performed using the prcomp function (centering and scaling data) from Stats R Package version 4.3.0, according to the expression level of genes, and plotted using ggplot2 R package version 3.4.2. The correlation among the samples was determined by using the adegenet library [24] of the statistical software package R (http://www.r-project.org, accessed on 1 January 2023) for their acceptance as biological replicates. Heatmaps were generated with a selection of bovine fibroblast DEGs in the pathways identified by using the heatmap.2 function from the gplots R package version 3.1.3.

#### 2.2.4. Functional Enrichment Analyses

The functional enrichment analyses to identify the enriched Gene Ontology (GO) terms (biological process (BP) and molecular function (MF)) and the Kyoto Encyclopedia of Genes and Genomes (KEGG) pathways were performed using the g:Profiler web server (https://biit.cs.ut.ee/gprofiler/gost, accessed on 10 February 2023) [17]. The g:SCS significance threshold was employed, which is the default method within g:Profiler for correcting multiple testing for *p*-values obtained from GO and KEGG pathway enrichment analysis. This threshold corresponds to an experiment-wide significance level of a = 0.05. Adjusted *p*-values were calculated by multiplying the *p*-values of the query by the ratio of the approximate threshold and the initial experiment-wide threshold (a = 0.05).

### 2.3. Analysis of Gene Expression in the Scrotal Skin of Naturally Infected Bulls

Scrotal skin samples obtained from naturally infected bulls were used to assess the expression of a set of genes chosen from the transcriptomic data. The scrotal skin was chosen due to the extensive occurrence of fibrosis in acutely and chronically infected bulls, as noted by González-Barrio et al. [4,5]. We selected genes that showed the most significant changes in the differential expression analysis and DEG representative of the main enriched KEGG pathways. The studied samples came from fifteen naturally infected breeding bulls from extensive beef herds (5 acutely infected bulls, 10 chronically infected bulls and 9 non-infected bulls). Acutely infected bulls showed fever and orchitis, and only one showed 2–3 tissue cysts per skin section [4]. Chronically infected bulls displayed skin lesions and were sterile, with testis atrophy with azoospermia [5]. Scrotal skin tissue samples were collected from each bull, and the samples were frozen at −80 °C until RNA extraction. Gene expression analyses were performed as previously explained [17]. Briefly, total RNA was purified by using a QIAGEN RNeasy Mini Kit (Qiagen, Hilden, Germany) following homogenization with a QIAshredder (Qiagen, Hilden, Germany). RNA concentration and purity were measured using a NanoPhotometer Classic (Implen, Munich, Germany), and RNA integrity was checked by agarose gel electrophoresis with GelRed staining (Biotium Inc., Fremont, CA, USA). Afterward, reverse transcription was performed using a SuperScript VILO cDNA Synthesis Kit (Invitrogen, Thermo Fisher Scientific, Waltham, MA, USA) and up to 2.5 μg of total RNA in a 20 μL reaction. Quantitative real-time PCRs were performed in 25 μL using 12.5 μL of Power SYBR PCR Master Mix (Applied Biosystems, Thermo Fisher Scientific, USA), 10 pmol of each primer and 5 μL of the diluted cDNA samples. The primers are listed in Appendix A. The reactions were performed in an ABI 7500 FAST Real-Time PCR System (Applied Biosystems, Foster City, MA, USA). Relative expression levels were calculated using the comparative method 2^−ΔΔCt^ [25] after normalization with the housekeeping gene *β-actin* [23,26]. For data analysis, the Kruskal–Wallis test followed by Dunn’s test was used, and *p* ≤ 0.05 was considered statistically significant.

## 3. Results

### 3.1. Quality Analysis and Mapping of RNA-Seq Data

All the samples included in the RNA-Seq study passed the quality checks, ensuring their suitability for subsequent analyses. The RNA integrity number (RIN) varied between 9.6 and 10 in all the samples. The sequencing process accounted for approximately 1 billion reads between all the samples. After alignment, an average of 70% of the high-quality reads mapped to the reference *Bos taurus* (Appendix A). Furthermore, all replicates from the same condition clustered together according to PCA, shown in Appendix A.

### 3.2. Transcriptional Responses of B. besnoiti-Infected Fibroblasts Highlight Key Cancer and Fibrosis Pathways

Differential expression analysis revealed a higher number of DEGs in FI-Bb compared to FI at 12 h p.i. versus 32 h p.i, with a total of 479 DEGs (287 upregulated in infected cells and 192 downregulated) and 280 DEGs (172 upregulated and 108 downregulated DEGs), respectively. Among these DEGs, 91 were DEGs at both time points. To investigate the underlying mechanisms during *B. besnoiti* infection in bovine fibroblasts, we performed functional enrichment analysis for GO and KEGG annotations (Table 1). When FI-Bb and FI were compared at both 12 and 32 h p.i., an enrichment in BP and MF related to cell proliferation and communication was revealed. Notably, the examination of KEGG pathways highlighted shared regulation at 12 and 32 h p.i. in several cancer-related pathways, namely “Proteoglycans in cancer”, “Pathways in cancer”, “Gastric cancer” and “MAPK signaling pathway” (Figure 1).

Specifically, at 12 h p.i., the comparison between FI-Bb and FI demonstrated an enrichment of BP and MF associated with cytokines and growth factor activity. The KEGG pathway analysis identified several pathways, including “Cytokine–cytokine receptor interaction, the “AGE–RAGE signaling pathway” and “TNF signaling pathway”. Notably, in infected cells, there was a significant upregulation of the cytokines and cytokine receptor (transforming growth factor β (TGFβ) family, TNF family, IFN family, chemokines family), vascular endothelial growth factors (*VEGFA*, *VEGFC*), the *fibroblast growth factor 1* (*FGF1*) and relevant genes associated to fibrosis such as *fibronectin type III domain-containing protein 3A* (*FNDC3A*), *collagen type VII A1* (*COL7A1)*, *matrix metalloproteinase 9* (*MMP9*) transforming growth factor-beta factors (*TGFβ1*, *TGFβ3*) and *urokinase plasminogen activator surface receptor* (*PLAUR*). Other genes associated with cell proliferation and differentiation were also upregulated (e.g., *NOTCH1*, *FOFB*, a member of the Fos family of AP-1 transcription factors and *phosphatidylinositol-specific phospholipase C*, *X domain containing 1* (*PLCXD1*) (Figure 2).

At 32 p.i., the comparison between FI-Bb and FI demonstrated an enrichment of processes related to cell adhesion. KEGG pathway analysis unveiled several pathways, including “Focal adhesion”, “Cell adhesion molecules, “PI3K–AKT signaling pathway” and the “Malaria” (HGF–MET signaling pathway). Some upregulated genes were also upregulated at 12 h p.i (FOSB, *PLCXD1*, *TGFβ1*). Particularly at 32 h p.i, there was a significant upregulation of several adhesion molecules (*CDH2*, *CLDN15*, *ITGA2*, *ITGA5*, *ITGA6*, *LRRC4*, *SDC4*, *PECAM1*), genes associated with fibrosis (*TGFβ1*, *TGFβ3*, *transforming growth factor β receptor 2—TGFβR2*—, *PLAUR*, *collagen type XIII A1—COL13A1*—, *matrix metallopeptidase 16* (*MMP16*) and *cell proliferation (FOSB*, *Amphiregulin*—*AREG*—) (Figure 2).

### 3.3. PLAUR, TGFβ1 and FOSB as Potential Biomarkers of B. besnoiti Infection in Scrotal Skin of Naturally Infected Bulls

The expression of a set of selected DEGs from the RNA-Seq data (*AREG*, *FGF1*, *FOSB*, *MMP9*, *MMP16*, *PLAUR*, *PLCXD1*, *TGFβ1*) was analyzed in scrotal skin samples from naturally infected bulls [4]. Among the eight genes examined, three showed distinct regulatory patterns in the scrotal skin of naturally infected bulls. Specifically, *PLAUR* exhibited upregulation in both acutely and chronically infected bulls when compared to the non-infected group. Moreover, *TGFβ1* was found to be upregulated in chronically infected bulls compared to the non-infected group. In contrast, only *FOSB* showed downregulation in chronically infected bulls. Notably, the remaining assessed genes (*FGF1*, *FGF1*, *MMP9*, *MMP16*, *AREG*, *PLCXD1*) did not display significant expression changes when infected and non-infected animals were compared (Figure 3).

## 4. Discussion

Primary bovine aorta fibroblasts were chosen as the in vitro model for this study due to their relevance as target cells for the parasite, as well as their intrinsic roles as immune modulators and pivotal contributors to fibrosis processes. It is noteworthy, however, that transcriptomic profiles may exhibit variability across fibroblasts of different origins. Fibroblasts represent a heterogenous cell population with diverse functional capacities, as exemplified by studies focused on skin-derived fibroblasts [27,28]. Nonetheless, our in vitro model holds promise as a valuable tool for studying fibrosis in the context of *B. besnoiti* infection, supported by the upregulation of classic fibrotic markers such as *TGFβ1*, *FNDC3A* and COL7A1.

Under physiological conditions, myofibroblasts are relatively rare. However, following infection and injury, fibroblasts typically undergo a transformation into myofibroblasts, serving as key drivers in the wound-healing process [29]. In our study, the upregulation of *TGFβ1*, *VEGFs* and *NOTCH1*—all known promoters of this fibroblast-to-myofibroblast transformation—was observed.

However, it is worth noting that the full transition to a myofibroblast phenotype generally involves the expression of alpha-smooth muscle actin (α-SMA) [30], which was not differentially expressed in *B. besnoiti*-infected fibroblasts. Therefore, it is possible that the observed changes represent an early step in myofibroblast transformation.

The differential expression analysis revealed a notably higher number of DEGs in FI-Bb vs. FI at 12 h p.i compared to 32 h p.i, suggesting an intensive early cellular reaction to the pathogen during invasion. However, the common core set of genes regulated at both time points showed initial steps of fibrosis, also observed at the host–tumor interface, evidenced by several enriched cancer-related pathways such as angiogenesis, apoptosis inhibition, cell proliferation and migration, which in turn related to innate immunity required for control and clearance of invading pathogens [31,32]. In addition, the MAPK pathway was upregulated at both times p.i., which is significant in relation to fibrosis, as it governs well-preserved cascades regulating cell proliferation, immune response, and inflammation [33].

Three phases are often presented sequentially during the fibrotic process: inflammation, proliferation and maturation [34]. Our transcriptomics findings suggest a gradual regulation of the fibrotic process. At 12 h p.i., pathways associated with the inflammatory response were enriched (e.g., “Cytokine–cytokine receptor interaction” “TNF signaling pathway” “AGE–RAGE signaling pathway”). Inflammation might facilitate the recruitment of immune cells, driven by the action of cytokines [35]. Nevertheless, inflammation could act as a double-edged sword; while it could effectively help to eliminate the invading parasite, prolonged activation could lead to tissue damage and, ultimately, fibrosis. Additionally, the upregulation of the “AGE–RAGE signaling pathway” could be contributing to this process by inducing oxidative stress and inflammation [36].

With parasite replication at 32 h p.i., the transcriptomic response evolves towards cell proliferation and ECM remodeling. The “PI3K–AKT signaling pathway” is well-known for its role in promoting cell survival and growth [37,38]. The process of tissue repair relies on fibroblasts’ ability to anchor themselves to ECM, which in turn leads to cell migration and ECM contraction. In fact, upregulation of adhesion signaling pathways is a key phenotypic hallmark of fibrotic cells [39]. While adhesion serves as a central feature of the proliferation phase, its significance perseveres and even amplifies as the fibrotic process advances into the maturation phase, since cell adhesion molecules are involved in binding to the extracellular matrix [40,41]. Moreover, the upregulation of the “Malaria” pathway, and more specifically, the HGF–MET signaling axis within this pathway, might promote a favorable environment for parasite infection by the inhibition of cell apoptosis [42]. Additionally, previous investigations have elucidated the role of the HGF–MET signaling pathway in inducing a hypermigratory phenotype in dendritic cells infected with *T. gondii* [43].

All these findings led us to identify potential fibrosis markers that displayed upregulation exclusively at 12 h p.i. (*FNDC3A*, *COL7A1*, *MMP9*) [44,45], at 32 h p.i. (*TGFβR2*, *COL13A1*, *MMP16*) [46,47] or at both time points (*PLAUR*, *TGFβ1*, *TGFβ3*) [48,49,50]. Among a set of selected and highly regulated DEGs involved in the different stages of the fibrotic process, *PLAUR*, *TGFβ1* and *FOSB* evidenced their potential as relevant biomarkers of disease progression in the scrotal skin of naturally infected bulls. *PLAUR* is involved in fibroblast-to-myofibroblast differentiation and fibrosis [51,52] and was upregulated in both acutely and chronically infected bulls. *TGFβ1* is intricately linked to the pathogenesis of fibrosis [53] and has been correlated with the activation of MAPK activity in fibroblasts during the fibrosis progression [54]. Herein, chronically infected bulls exhibited upregulation of *TGFβ1*, and the MAPK pathway was upregulated at both time points. In contrast, *FOSB* expression did not correlate between the in vitro and in vivo assays. *FOSB* was downregulated in the scrotal skin of chronically infected bulls but was upregulated in vitro at both times p.i. Downregulated expression of *FOSB* has been associated with cell migration (metastasis) and advanced tumorigenic stages [55] whereas its upregulation is associated with regulation of ECM production. Consequently, tissue complexity and infection timing likely contributed to differences between the in vitro and in vivo models and to the variability observed in the mRNA expression levels in the chronically infected group.

It is not surprising that “Pathways in cancer” was enriched at both time points studied since cancer-associated fibrosis is a critical component of the tumor microenvironment that correlates with prognosis. Moreover, tumors are characterized by ECM deposition, remodeling and angiogenesis, which are fibrosis steps that are regulated by *B. besnoiti* [56]. Herein, we also found a relevant DEG at both time points studied that is a marker of angiogenesis, VEGFA. VEGF signaling results in the activation of several downstream pathways including the RAS/MAPK (ERK) pathway regulating cell proliferation and the PI3K–AKT pathway, regulating cell survival, among others, that also were found enriched in this work (Table 1). A functional in vitro study carried out with an ERK inhibitor demonstrated that ERK inhibition at 12 h p.i. may prevent endothelial sprouting by downregulating VEGFA expression (Appendix A). It has also been reported that VEGF autoregulates ERK1/2 and p38 cascades enhancing the expression of DUSP phosphatases (MAPK phosphatases) [57]. Interestingly, DUSP5 and DUSP6 were also found to be upregulated at 12 h p.i. (Figure 2).

Similar transcriptomic studies have been developed in other primary bovine cell targets of *B. besnoiti* infection (bovine aorta endothelial cells (BAEC) and macrophages), and some regulated genes and pathways related to immune response and fibrosis were common to all the cell lines. Regarding the fibrotic process, the TGFβ signaling pathway was commonly upregulated. The antiviral immune response was represented by several DEGs (*OAS1Z*, *IFI44*, *PIM*, *IL7R*, *CISH*) and the JAK–STAT pathway. Although *OAS1Z* and *IFI44* are recognized for their roles in viral infections [58,59], they can also be induced in response to certain parasites [60,61,62]. In addition, the JAK–STAT pathway is known for orchestrating potent antiviral reactions by leading to the upregulation of numerous interferon-stimulated genes (ISGs), which can be also triggered by apicomplexan parasites as *T. gondii* [63] and *N. caninum* [64], and, indeed, the role of ISGs was demonstrated during *B. besnoiti* infection in primary bovine macrophages [17]. Moreover, genes associated with programmed cell death were also regulated in all bovine cells as well, which could favor the infection, as shown for *T. gondii* [65]. In addition, previous studies have described apoptosis as one of the main pathways regulated in primary bovine macrophages during early infection [64]. Additionally, there were also commonly regulated adhesion molecules that serve a crucial role in the host’s defense against infections, facilitating processes such as pathogen recognition [66], pathogen internalization, cytoskeletal rearrangements, and even influencing gene expression events that ultimately impact the phenotype of the infected cell [67].

## 5. Conclusions

Our study sheds light on the molecular mechanisms governing fibroblast activation during infection, revealing a progressive modulation of fibrotic processes that share hallmarks with cancer progression. MAPK signaling arises as a key element together with other relevant pathways (e.g., TNF and cell adhesion) whose role in bovine pathogenesis remains to be dissected. Moreover, several potential biomarkers of the different phases of fibrosis, including *PLAUR*, *TGFβ1* and *FOSB*, which were also regulated in vivo, were identified. The DEGs and associated pathways identified in infected fibroblasts may serve as invaluable prognostic indicators and potential drug targets for combating fibrosis triggered by *B. besnoiti* infection (e.g., anti-angiogenic therapies [68,69,70]). This study represents a first attempt to identify putative biomarkers of fibrosis with the most appropriate samples currently available from bovines with besnoitiosis. These results should be further validated with complementary methodologies (e.g., Western blot or ELISA), which will also require additional samples from the field.

## Figures and Tables

**Figure 1 microorganisms-12-00586-f001:**
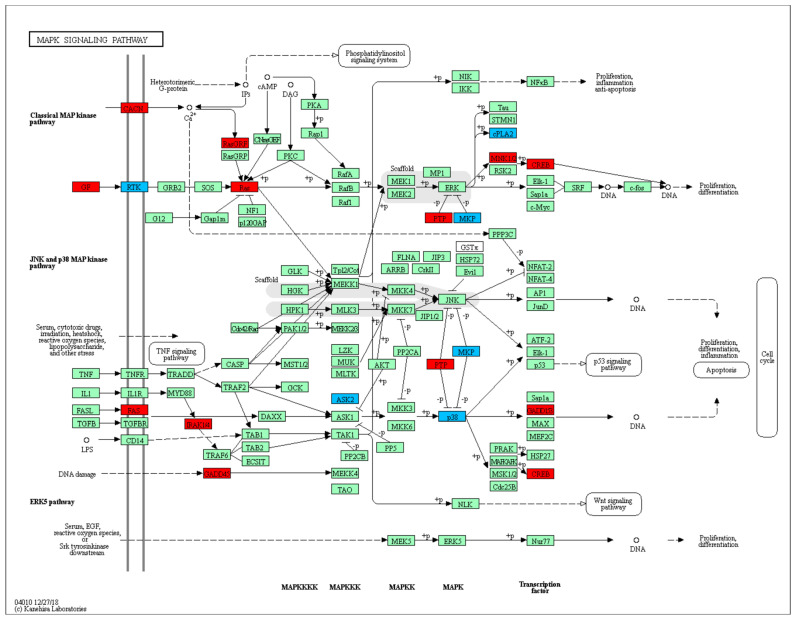
MAPK signaling pathway (KEGG: 04010) enriched in bovine fibroblasts infected with *B. besnoiti*. DEGs at 12 h p.i. is represented in blue and DEGs at 32 h p.i. in red.

**Figure 2 microorganisms-12-00586-f002:**
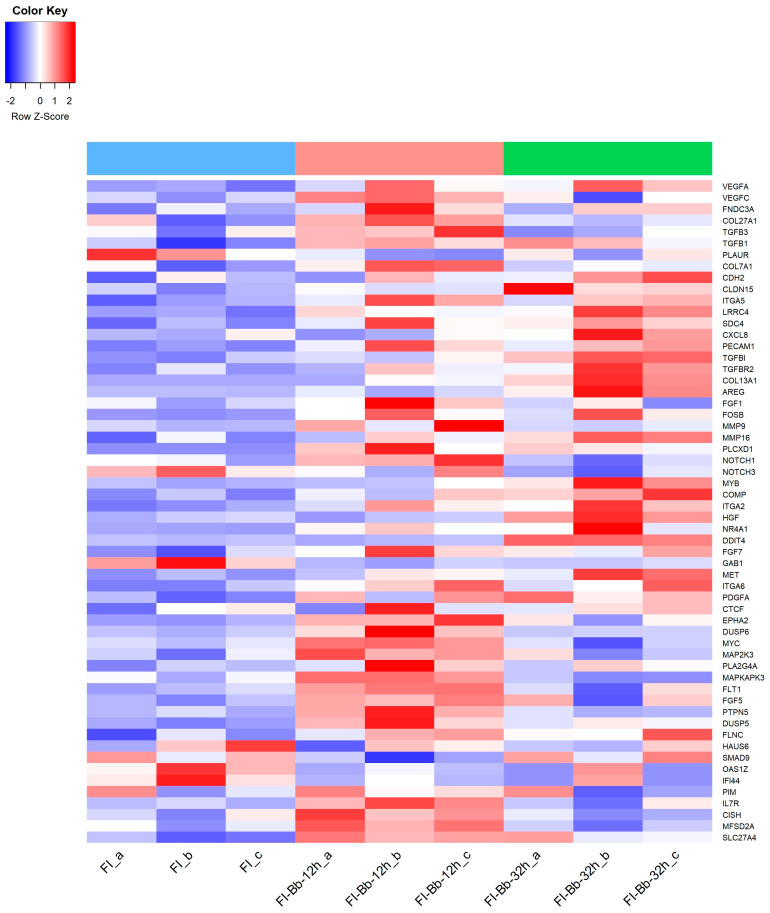
Heatmap of a selection of *Bos taurus* differentially expressed genes (DEGs) in fibroblasts infected with *B. besnoitia* tachyzoites (FI-Bb) at 12 (pink color) and 32 h p.i. (green color) and non-infected fibroblasts (FI) (blue color).

**Figure 3 microorganisms-12-00586-f003:**
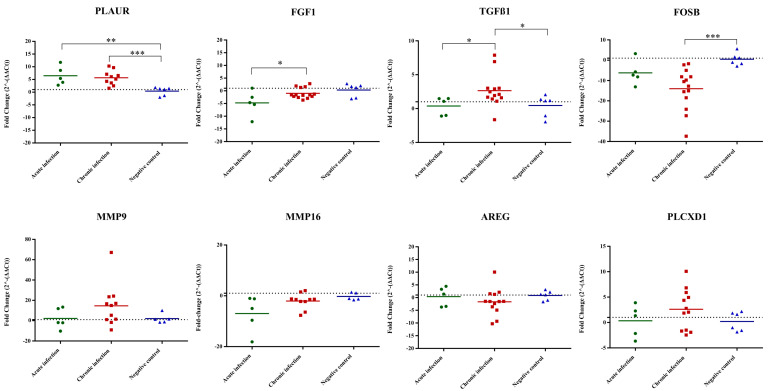
Relative mRNA expression levels of selected DEGs determined by quantitative real-time PCR (RT-qPCR) analysis of scrotal skin samples from naturally infected bulls. In the scatterplot graphs, each dot represents data obtained from one single bull, and the bar represents the average fold change (FC) for each gene and condition. The baseline for uninfected animals is set at 1 (horizontal line). One asterisk (*) corresponds to adjusted *p* values between 0.01 and 0.05; two asterisks (**) to adjusted *p* values between 0.01 and 0.001; and three asterisks (***) to adjusted *p* values less than 0.001 (determined by Kruskal–Wallis test followed by Dunn’s test).

**Table 1 microorganisms-12-00586-t001:** GO and KEGG enrichment analyses. The number of differentially expressed genes (DEGs) between fibroblasts infected with *Besnoitia besnoiti* tachyzoites (FI-Bb) and non-infected fibroblasts (FI) are shown together with the top 6 upregulated Gene Ontology (GO) terms, biological processes (BP) and molecular functions (MF), and top 5 upregulated Kyoto Encyclopedia of Genes and Genomes (KEGG) pathways).

	No. DEGs	Upregulated DEGs	Downregulated DEGs	Biological Processes		Molecular Function		Pathway Enrichment	
				GO Term	Adjusted *p*-Value	GO Term	Adjusted *p*-Value	KEGG Term	Adjusted *p*-Value
**FI-Bb vs. FI 12 h p.i.**.	479	287	192	Positive regulation of biological process (GO:0048518)	2.60 × 10^−18^	Molecular function regulator activity (GO:0098772)	8.70 × 10^−8^	Cytokine–cytokine receptor interaction (KEGG:04060)	3.92 × 10^−4^
				Anatomical structure development (GO:48856)	2.43 × 10^−17^	Molecular function activator activity (GO:0140677)	9.15 × 10^−8^	Pathways in cancer (KEGG:05200)	3.94 × 10^−4^
				Anatomical structure morphogenesis (GO:0009653)	6.64 × 10^−17^	Protein binding (GO:0005515)	6.01 × 10^−6^	MAPK signaling pathway (KEGG:04010)	1.32 × 10^−3^
				Developmental process (GO:0032502)	1.67 × 10^−16^	Signaling receptor regulator activity (GO:0030545)	6.07 × 10^−5^	Proteoglycans in cancer (KEGG:05205)	5.32 × 10^−3^
				Positive regulation of cellular process (GO:0048522)	6.89 × 10^−16^	Receptor ligand activity (GO:0048018)	1.96 × 10^−4^	Axon guidance (KEGG:04360)	3.13 × 10^−2^
				Multicellular organism development (GO:0007275)	9.70 × 10^−15^	Signaling receptor activator activity (GO:0030546)	3.03 × 10^−4^	TNF signaling pathway	3.43 × 10^−2^
**FI-Bb vs. FI 32 h p.i.**	280	172	108	Multicellular organism development (GO:0007275)	1.28 × 10^−16^	Protein binding (GO:0005515)	3.47 × 10^−6^	Proteoglycans in cancer (KEGG:05205)	4.18 × 10^−5^
				Anatomical structure development (GO:0048856)	8.42 × 10^−15^	Identical protein binding (GO:0042802)	7.88 × 10^−5^	Malaria (KEGG:05144)	3.59 × 10^−3^
				System development (GO:0048731)	1.22 × 10^−14^	Signaling receptor binding (GO:000510)	3.95 × 10^−4^	Focal adhesion (KEGG:04510)	7.48 × 10^−3^
				Developmental process (GO:0032502)	1.88 × 10^−14^	Collagen binding (GO:0005518)	1.55 × 10^−3^	Pathways in cancer (KEGG:05200)	7.51 × 10^−3^
				Anatomical structure morphogenesis (GO:0009653)	4.38 × 10^−14^	Signaling receptor regulator activity (GO:0030545)	3.63 × 10^−3^	MAPK signaling pathway (KEGG:04010)	1.08 × 10^−2^
				Multicellular organismal process (GO:0032501)	1.99 × 10^−12^	Receptor ligand activity (GO:0048018)	5.99 × 10^−3^	PI3K–AKT signaling pathway (KEGG: 04151)	2.89 × 10^−2^

## Data Availability

The data are contained within the article or Appendix A. Additionally, the RNA-Seq data reported in this study have been deposited in the NCBI database under accession number SUB14163821.

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
