# Peer review of "Transcriptomics of Besnoitia besnoiti-Infected Fibroblasts Reveals Hallmarks of Early Fibrosis and Cancer Progression"

_microorganisms, 2024, doi:10.3390/microorganisms12030586_

Round 1

Reviewer 1 Report

Comments and Suggestions for Authors

The results of the study are informative and comprehensive. The work is scientifically meaningful. However, there are some weakness and should be addressed before it can be accepted for publication.

Major comments

1、  In Fig3. TGFß1 can only predict chronic infection, cannot discriminate acute infection and negative control samples, while other pathogens (bacteria, virus and other parasites) may also regulate the expression of the gene. The prognostic significance of the biomarker need further study.

2、  In Fig3. PLAUR can discriminate the infected samples with uninfected. It seems like a good biomarker to help diagnosis. But it cannot discriminate acute infection and chronic infection.

3、  In Fig3. FOSB cannot discriminate acute infection and negative control samples, and FOSB expression did not correlate between the in vitro and in vivo assays. Based on the data, it is not a suitable biomarker to anticipate the process of the disease.

4、  In Fig3. The Variations in the data from chronic infection group is large, please give some reasons on the issue. If it is related to different clinical stages, different sampling sites. How to set up quality control should be addressed.

5、  The expression level on gene and protein level may not consist, the gene could be regulated at epigenetic level. The potential biomarkers and players in host pathways should be evaluated in both levels. In this study, only RNA seq, no western blot data, the authors should give some explanations.

6、  In the abstract, the conclusion should be more clarity. Which biomarker is the best?

Author Response

We appreciate these valuable remarks, and we hope to better clarify this issue. It has been reported that gene expression results may differ between in vitro and in vivo models of infection due to dynamic gene regulation influenced by cell types and environmental conditions (1-2). Moreover, an additional limitation of working with samples form naturally infected animals is the individual variability of results influenced by age and non-synchronised infection status. Accordingly, it is difficult to extrapolate results obtained in a homogeneous in vitro fibroblast cell culture to a complex tissue collected from naturally infected animals. In addition, a non-synchronised infection status might explain the variability observed in the chronically infected group. We have included this sentence in lines 443-44: “and to the variability observed in the mRNA expression levels in the chronically infected group”.

Despite these limitations, these valuable samples obtained from culled breeding bulls constituted the best target samples as they had been previously characterized by histopathological and molecular methods. Western blotting provides valuable protein-level data. However, the correlation between transcriptomic and proteomic data is not always direct, due to post-transcriptional and post-translational modifications. Accordingly, this study represents a first attempt to identify putative biomarkers of fibrosis with the most appropriate samples currently available from bovines with besnoitiosis. However, we agree with the reviewer that these results should be further validated with complementary methodologies (eg. western blot or ELISA) that will also require additional samples from the field.  Ideally, a longitudinal study conducted in an experimental model of infection should be conducted. Unfortunately, there is not an appropriate bovine model of chronic infection. Diezma-Diaz et al., developed a model of chronic besnoitiosis in calves but there are serious limitations that hamper its reproducibility as it is based on the intradermal inoculation of bradyzoites isolated from naturally infected animals. Based on all these arguments, the present results do not permit to give conclusive results regarding the best biomarker. At a first glance, PLAUR and TGFß1 appear to be the most promising biomarkers of fibrosis regardless the stage of the infection, but FOSB together with other DEGs such as MMP9, MMP16, AREG, and PLCXD1should not be ruled out. We have tried to give more emphasis on this issue, and we added this sentence in the conclusion: This study represents a first attempt to identify putative biomarkers of fibrosis with the most appropriate samples currently available from bovines with besnoitiosis. These results should be further validated with complementary methodologies (eg. western blot or ELISA) that will also require additional samples from the field.

Reviewer 2 Report

Comments and Suggestions for Authors

All comments are in the revised pdf

Comments on the Quality of English Language

A minor English edition is required.

Punctuation should be revised.

Author Response

We have included all the suggestions made by the reviewer with three exceptions:

  • Page 3, section 2.2.3: Oligonucleotide primer pairs should be mentioned or listed in a supplementary table. All oligonucleotide pairs employed in this study were already listed in Supplementary table 1 cited in section 2.3.
  • Page 12. Do not capitalize “Pathways in cancer”. We have maintained the capital letter since “Pathways in cancer” is the KEGG term (https://www.genome.jp/pathway/hsa05200) and all the KEGGs pathways were written with capital letters along the manuscript.
  • Conclusion is the section that summarizes my results, so no references are required: we have maintained the cited references since they support the example added between brackets.

Finally, following the remark made by the reviewer (The figure quality (Fig.3) should be improved) the figure has been enlarged and the original good quality file was also submitted.

Reviewer 3 Report

Comments and Suggestions for Authors

The study tries to decipher the molecular mechanisms that are involved in the activation of fibroblasts in Besnoitia besnoiti infection. The authors identified a progressive modulation of fibrotic processes, which have distinctive signs with cancer progression. 

In addition, several potential biomarkers of different stages of fibrosis have been identified in vivo. 

DEGs and associated pathways found in infected fibroblasts may represents prognostic indicators and potential drug targets to combat the fibrosis encountered in B. besnoiti infection.

Identifying MAPK opens the way for new research directions.

All the considerations highlight the originality the research carried out and presented in this study. 

The methodology is appropriate of the objectives pursued.

The conclusions are synthetic and refer to the problem that this study wants to solve. 

The bibliographic references are well chosen and many very recent, which proves that the subject is current.

 The table and figures presented in this study are self-explanatory and complement this study favorably.

It is an excellent study that complements data from the specialized literature. It think it can be published in this form. 

Author Response

We appreciate so much your positive comments.

Reviewer 4 Report

Comments and Suggestions for Authors

An interesting manuscript describing gene expression in Besnoitia-infected fibroblasts with the hope of identifying points on intervention in cattle infections. The manuscript is well-written and the discussion thorough.  

Section 2.1 what temperature and CO2 concentration were used to culture the cells and parasites?

L98 - with what were the cultures washed?

L177 - it is not a good practice to use Actin as the housekeeping gene in conditions that contribute to structural changes (cytoskeletal rearrangements - L421). A gene such as GAPDH would have been a more appropriate choice. 

Figure 3. It is nearly impossible to see the individual graphs and text at this size. Please enlarge. Also, * and ** are described but *** also appears - what does it represent?

Please update acknowledgements and pagination.

Comments on the Quality of English Language

In general the manuscript is very well-written, but in some places the grammar is incorrect and the use of terms such as "end up with" should be changed to the more appropriate "results in". This is not an inclusive list, the entire manuscript should be reviewed.

Author Response

An interesting manuscript describing gene expression in Besnoitia-infected fibroblasts with the hope of identifying points on intervention in cattle infections. The manuscript is well-written and the discussion thorough. 

Section 2.1 What temperature and CO2 concentration were used to culture the cells and parasites?

Cell culture incubator conditions were 37ºC and 5% of CO2. This information has been added in line 99.

L98 - With what were the cultures washed?

Cultures were washed with phosphate-buffered saline 1x. This information has been added in line 107.

L177 - it is not a good practice to use Actin as the housekeeping gene in conditions that contribute to structural changes (cytoskeletal rearrangements - L421). A gene such as GAPDH would have been a more appropriate choice.

We agree with the reviewer's comment. We considered Actin as an appropriate housekeeping gene according to our previous results in which Actin and GAPDH were used as housekeeping genes in different cells infected by apicomplexan parasites and differences between them were not found (3-6). However, following the reviewer´s suggestion we will not make this assumption in future experiments and at least two different housekeeping genes will be used. 

Figure 3. It is nearly impossible to see the individual graphs and text at this size. Please enlarge. Also, * and ** are described but *** also appears - what does it represent?

We have already enlarged Figure 3. In caption to figure you will find the explanation to all the symbols: One asterisk (*) corresponds to adjusted p values between 0.01 and 0.05, two asterisks (**) to adjusted p values between 0.01 and 0.001 and three asterisks (***) to adjusted p values less than 0.001 (by Kruskal‒Wallis test followed by Dunn’s test).

Please, update acknowledgements and pagination.

Acknowledgements section has been updated and pagination format depends on the journal.

Round 2

Reviewer 1 Report

Comments and Suggestions for Authors

The  modified version is OK. It could be published in the magazine.